# Advances for Managing Pancreatic Cystic Lesions: Integrating Imaging and AI Innovations

**DOI:** 10.3390/cancers16244268

**Published:** 2024-12-22

**Authors:** Deniz Seyithanoglu, Gorkem Durak, Elif Keles, Alpay Medetalibeyoglu, Ziliang Hong, Zheyuan Zhang, Yavuz B. Taktak, Timurhan Cebeci, Pallavi Tiwari, Yuri S. Velichko, Cemal Yazici, Temel Tirkes, Frank H. Miller, Rajesh N. Keswani, Concetto Spampinato, Michael B. Wallace, Ulas Bagci

**Affiliations:** 1Machine and Hybrid Intelligence Lab, Feinberg School of Medicine, Northwestern University, Chicago, IL 60611, USA; deniz.seyithanoglu@northwestern.edu (D.S.); gorkem.durak@northwestern.edu (G.D.); elif.keles@northwestern.edu (E.K.); alpay.m@istanbul.edu.tr (A.M.); zilianghong2029@u.northwestern.edu (Z.H.); zheyuan.zhang@northwestern.edu (Z.Z.); y-velichko@northwestern.edu (Y.S.V.); frank.miller@nm.org (F.H.M.); raj-keswani@northwestern.edu (R.N.K.); 2Istanbul Faculty of Medicine, Istanbul University, Istanbul 38000, Turkey; yavuztaktak@istanbul.edu.tr (Y.B.T.); timurhan.cebeci@istanbul.edu.tr (T.C.); 3Department of Radiology, BME, University of Wisconsin-Madison, Madison, WI 53707, USA; ptiwari9@wisc.edu; 4William S. Middleton Memorial Veterans Affairs (VA) Healthcare, 2500 Overlook Terrace, Madison, WI 53705, USA; 5Department of Gastroenterology, University of Illinois at Chicago, Chicago, IL 60611, USA; cyazic2@uic.edu; 6Department of Radiology, Indiana University, Indianapolis, IN 46202, USA; atirkes@iupui.edu; 7Department of Electrical, Electronics and Computer Engineering, University of Catania, 95124 Catania, Italy; concetto.spampinato@unict.it; 8Department of Gastroenterology, Mayo Clinic Florida, Jacksonville, FL 32224, USA; wallace.michael@mayo.edu

**Keywords:** pancreatic cystic lesions, IPMN, precancerous cysts, pancreas imaging, AI for pancreatic diseases, pancreatic cancer

## Abstract

Pancreatic cystic lesions (PCLs) can range from harmless growths to precursors of pancreatic cancer, making accurate diagnosis crucial for patient care. Traditional methods for managing PCLs, such as imaging and biopsies, often depend on the skill of the doctor interpreting the images, leading to variability in diagnosis and treatment. This review highlights the challenges in diagnosing and managing PCLs and discusses the potential for artificial intelligence (AI) to improve accuracy. AI techniques, such as automated image analysis and deep learning algorithms, can provide more consistent and reliable assessments of pancreatic cysts. These tools could help doctors better identify high-risk lesions that need treatment, while avoiding unnecessary procedures for benign cysts. AI-driven methods show promise in improving patient outcomes, offering earlier detection and more precise management, and ultimately helping to prevent pancreatic cancer.

## 1. Introduction

Pancreatic cancer, a serious malignancy that develops in the tissues of the pancreas, is a growing concern due to its rising incidence and relative lack of improved survival compared with other common cancers. It has recently become the third leading cause of cancer-related deaths in the United States, with overall 5-year relative survival rate (between 2014 and 2020) for all stages remaining low at 12.8% [1].

Globally, pancreatic cancer ranks seventh among cancer-related deaths, and it is expected to rise to the second leading cause of cancer deaths in Western countries [2]. While increased life expectancy and aging have been linked to rising pancreatic cancer incidence [3], the latest data has shown a disturbing trend: younger people, particularly women, are experiencing a rise in pancreatic cancer diagnoses [4,5]. This increase is likely to be due, in part, to growing rates of obesity, diabetes, smoking, and alcohol consumption [2].

Pancreatic ductal adenocarcinoma (PDAC) is the most common type of pancreatic cancer, making up about 90% of all cases [2]. Although the lifetime incidence of PDAC is relatively low at 1.7%, most patients are diagnosed at advanced stages [1]. This means that only 15–20% of patients have surgically resectable disease at the time of diagnosis [2,6].

### 1.1. Challenges in Early Diagnosis of PDAC

Patient life expectancy and outcomes differ significantly depending on the stage of PDAC. For patients with stage I PDAC, which is surgically resectable, the long-term survival rate is around 80% [7]. Unfortunately, diagnosis at this stage is uncommon, since 75–80% of PDACs develop from pancreatic intraepithelial neoplasia (PanIN), which are precursor lesions that cannot be detected with current standard imaging methods, requiring biopsy and microscopic examination for definitive diagnosis [8,9,10]. The only available opportunity to detect potential pancreatic cancer at earlier stages is through pancreatic cystic lesions (PCLs), which can be visualized with current imaging methods [11].

A proportion of 15–20% of PDACs develop from PCLs, which are increasingly detected due to aging populations, increased prevalence of risk factors, more frequent use of imaging methods, and advancements in these imaging techniques [11,12]. Recent studies show that the incidence of PCLs is between 1.2 and 2.6% on computed tomography (CT), reaching up to 49.1% on magnetic resonance imaging (MRI) [12]. The prevalence of PCLs in the general population is estimated to be in the range of 4–14%, with autopsy studies demonstrating a prevalence reaching 50% in aging populations [13,14]. Current guidelines do not recommend routine pancreas screening for PCLs. Only a small fraction of PCLs progress to PDAC [2].

Traditional challenges in imaging of PCLs include difficulty in differentiation between benign and malignant cysts based solely on imaging features, the inconsistent appearance of cysts across different types, and the potential for small, subtle features that could be missed, all of which can lead to uncertainty in management decisions [15,16,17]. Often, these challenges necessitate additional invasive procedures, like endoscopic ultrasound (EUS) with fine-needle aspiration (EUS–FNA), and surgical intervention, which are expensive and carry significant morbidity and mortality risks [9,18,19]. The challenge lies in accurately identifying which PCLs pose a real threat and necessitate early intervention, while simultaneously avoiding interventions for lesions that are unlikely to cause problems.

### 1.2. The Potential Role of Artificial Intelligence (AI) in Pancreatic Diseases

Given the potential for long-term survival with early cancer diagnosis, it is critical to develop more accurate methods to distinguish early precursors of malignancy. Artificial intelligence (AI), particularly deep learning (DL), is rapidly transforming the medical field [20], with significant advancements in pancreatic cancer research [21]. This area has historically been challenging due to late-stage diagnoses and the complex characteristics of the pancreas and its associated pathologies, like solid tumors and cysts [22]. AI offers the ability to analyze vast numbers of medical images, like CT and MRI scans, and electronic health records (EHRs) to make reliable diagnoses and predictions [23]. DL and radiomics, two emerging approaches in AI, show strong potential for improving the utility of imaging, clinical, and EHR data in pancreatic cancer diagnosis [24]. For example, diagnosing suspicious lesions, whether solid tumors or pre-cancerous cysts, typically involves several steps. Figure 1 provides an overview of an AI-based workflow for pancreatic diseases (cancer and others). First, raw medical images undergo preprocessing to enhance their quality and prepare them for standardized analysis. A key step is to segment the pancreas and the suspected regions, isolating them for further targeted analysis. Segmentation remains a major challenge, especially for the pancreas, but it is essential for developing automated diagnostic and prognostic applications. After segmentation, the isolated pancreas images undergo radiomics and/or DL analysis for downstream evaluations. The final step involves presenting the AI-derived results to clinicians, who may use this information to make ultimate decisions about treatment, including surveillance recommendations, surgical planning, personalized treatment strategies, and response assessments.

AI offers numerous advantages for improving patient management and diagnostic workflows in pancreatic diseases. For example, current clinical guidelines for pancreatic cyst management primarily rely heavily on cyst size, the presence of mural nodules, and dilation of the main pancreatic duct. However, these criteria often lack sensitivity and specificity, leading both to missed cancers and unnecessary surgeries [25]. High-risk PCLs present a critical opportunity for early detection of pancreatic cancer. AI holds great promise for improving the risk stratification of PCLs. By leveraging AI, it may be possible to predict which cysts harbor or will likely progress into aggressive tumors, both in the short and long term. This approach could significantly improve diagnostic accuracy, reduce the needs for invasive sampling and serial imaging, and enable more targeted, effective treatment strategies. Ultimately, it could reduce unnecessary interventions and improve patient outcomes.

This review aims to provide a comprehensive understanding of pancreatic cystic lesions, current patient management guidelines, and the application of AI in the radiologic assessment of PCLs with malignant potential. The societal impact of this review extends far beyond academic circles, offering substantial benefits to both healthcare delivery and patient outcomes. At its core, the review addresses a critical challenge in modern medicine—the accurate diagnosis and management of PCLs. By examining the potential of AI in this field, the review paves the way for more precise and standardized care delivery, potentially saving lives through earlier detection of pancreatic cancer while simultaneously reducing unnecessary procedures for patients with benign cysts. From an industry perspective, this review serves as a valuable roadmap for technological innovation and healthcare improvement. It creates new opportunities for the medical technology sector to develop and implement AI-based diagnostic tools, while providing healthcare providers with crucial insights for integrating these advanced technologies into clinical practice. The review’s comprehensive analysis of current limitations and future possibilities guides research and development efforts, potentially catalyzing collaborations between healthcare providers and technology companies. This could lead to the creation of more sophisticated diagnostic tools and standardized protocols for managing pancreatic cystic lesions.

This review is organized into the following sections: Pancreatic Cystic Lesions, Current Guidelines for PCL Management, Imaging Tests for PCL Evaluation, Artificial Intelligence for PCL Evaluation, Discussion and Future Prospects, and Conclusions.

## 2. Pancreatic Cystic Lesions (PCLs)

PCLs represent a heterogeneous group with varying properties and malignancy risks (See Figure 2) [26]. While there are more than 20 defined PCL types, there are six most common types, which fall into two primary categories: non-mucinous and mucinous [10,27]. Non-mucinous cysts include pseudocysts, serous cystic neoplasms (SCNs), solid-pseudopapillary neoplasms (SPNs) and cystic pancreatic neuroendocrine tumors (cNETs) [26]. Pseudocysts and SCNs make up 15–25% of all pancreatic cysts and, since both lesions are not thought to have a risk of malignant transformation, they usually do not require surveillance or intervention in asymptomatic patients [26,28]. SPNs and cNETs have moderate malignancy risks, albeit less than mucinous cysts [9,26]. Due to their malignant potential, surgical resection of SPNs and cNETs is generally recommended [26]. However, these patients have higher survival rates than PDAC patients [26]. Mucinous cysts are further divided into intraductal papillary mucinous neoplasms (IPMNs) and mucinous cystic neoplasms (MCNs), which make up around 50% of PCLs found incidentally on imaging, both having high-risk of progression to PDAC [26,27]. Currently, PCLs are differentiated using cross-sectional imaging techniques, such as CT and MRI, as well as EUS–FNA of cyst fluid [18]. Understanding the unique properties of each type of PCL is crucial for stratifying pancreatic cysts based on their potential to become malignant. Figure 2 provides a visual representation of the malignancy risk and broad classification of various PCLs [26,27].

### 2.1. Clinical and Histopathologic Properties of PCLs

**Pseudocysts** are complications of acute or chronic pancreatitis that form when debris, inflammatory cells, and blood accumulate within a thick fibrous wall, forming single or multiple cysts [27]. Unlike true cysts, they lack a lining of epithelial cells, hence the “pseudo” prefix [26,29]. It is important to distinguish that, while pseudocysts are a consequence of pancreatitis without malignant potential, new-onset pancreatitis with an incidental cyst warrant investigation for malignant cysts [27].

**Serous cystic neoplasms (SCNs),** also known as serous cystadenomas, are predominantly seen in women (around 75%), between the ages of 40 and 60 [29]. They typically consist of multiple microcysts that are lined by a single layer of cuboidal epithelium [29]. Although most SCNs are benign, asymptomatic lesions with slow growth, larger SCNs may cause symptoms like abdominal pain, and may cause pancreatitis and biliary obstruction warranting investigation [27].

**Cystic pancreatic neuroendocrine tumors (cNET)** account for about 15% of all pancreatic neuroendocrine tumors, and affect both men and women equally, typically developing between the ages of 30 and 50 [26]. They can occur sporadically or be associated with multiple endocrine neoplasia type 1 [30]. cNETs larger than 2 cm carry a moderate risk of malignancy and often require surgical removal [26].

**Solid-pseudopapillary neoplasms (SPNs)** are cysts that are lined by monomorphic cuboidal cells and have a fibrous pseudo-capsule [29]. They primarily affect young women (being around 10 times more common in women than in men) in their 20s but can occur at any age [29,30,31]. Due to their moderate malignancy risks, surgical removal is often performed, and long-term outcomes are excellent [26].

**Mucinous cystic neoplasms (MCNs)** are characterized by a lining of tall, columnar epithelium with surrounding ovarian-type stroma, resulting in being almost exclusively seen in women, usually aged between their 40s and 60s [29,32,33,34,35]. While MCNs carry a 10–17% risk of progression to PDAC, recent studies suggest that this risk may be lower for lesions smaller than 3 cm in diameter and lacking other high-risk features [32,33,35,36,37,38,39].

**Intraductal papillary mucinous neoplasms (IPMNs)** are the most common type of pancreatic cysts, comprising approximately half of all PCLs [11]. These cysts affect men and women equally and can develop at any age, although they are most frequently diagnosed in individuals between the ages of 40 and 60 [32]. Histopathologically, IPMNs are characterized by papillary growth of columnar neoplastic cells that produce increased amounts of mucin, which leads to pancreatic duct dilatation and obstructive jaundice [12,26]. Radiologically, IPMNs are classified according to their region-of-involvement as main pancreatic duct (MD-IPMN), branch duct (BD-IPMN), or both mixed (from both main and branch ducts) IPMNs [26,32].

BD-IPMNs are the most common subtype, with recent studies indicating that around 80% of incidentally discovered cysts are BD-IPMN [40,41]. In contrast, MD-IPMNs and mixed IPMNs account for 15–21% of all IPMNs [42]. MD-IPMNs have the highest risk of progressing to PDAC, although some BD-IPMNs with high-risk features also present a significant malignancy risk [30,32,43,44].

### 2.2. Radiological Properties of PCLs

PCLs have certain radiological properties that have been defined with current imaging methods. While these properties aid in differentiating between types, PCLs are not always easy to distinguish, even with EUS–FNA. While established radiological properties are discussed in this section, high-risk features are discussed in a further related section. Figure 3 illustrates examples of PCLs visible on both CT and MRI.

Pseudocysts appear as well-circumscribed, unilocular or multilocular cysts that usually have a connection to the main pancreatic duct [26,27]. This connection leads to elevated cyst fluid amylase levels on EUS–FNA cyst fluid sampling; on the other hand, they have low cyst fluid carcinoembryonic antigen (CEA) levels [19,29,30].

SCNs usually appear as multiple small cysts arranged in a “honeycomb” pattern (microcystic), but some may present as large cysts (macro-cystic), as unilocular lesions or as solid lesions [27]. In up to 30% of SCNs, there might be a central scar that is characteristic of SCNs [29]. cNETs appear as well-circumscribed solid, cystic, or mixed lesions, often located in the head of the pancreas, with a characteristic peripheral enhancing rim on imaging [27,29,30]. SPNs appear as a mix of solid and cystic components that could appear anywhere on the pancreas [26,29]. Smaller SPNs tend to be solid, while larger lesions contain more cystic and hemorrhagic components [29]. Distinguishing these cysts is challenging if they lack their characteristic lesions, even with EUS–FNA, since cyst fluid analysis of SCNs, cNETs and SPNs show low levels of amylase and CEA [26]. In cNETs, microscopic examination of the fluid could be useful, and it may reveal loosely cohesive round cells, characteristic of neuroendocrine tumors [26]. Furthermore, the majority of cNETs express somatostatin receptors, which can be visualized with positron-emission tomography using nuclear tracers, like octreotide or dotatate, aiding with diagnosis [27].

MCNs are seen on the body or tail of the pancreas 90–95% of the time, usually as single macro-cystic lesions with thick walls, some containing characteristic peripheral “eggshell” calcifications [26,27,30].

IPMNs are typically seen on the head of the pancreas and may present as single or multifocal lesions involving the main pancreatic duct (MD-IPMN), the branch duct (BD-IPMN), or both main and branch ducts (mixed IPMN) [26,32]. BD-IPMNs, the most common subtype, can present as a small “grape-like” cluster of cysts, while MD-IPMNs cause diffuse or segmental pancreatic duct dilatation [27]. In some MD-IPMNs, the dilated main pancreatic duct can extrude mucin into the duodenum, producing a pathognomonic endoscopic finding known as the “fish-mouth” appearance [26,27].

While cyst fluid analysis of MCNs shows low amylase levels, IPMNs have elevated amylase levels due to their connection with the pancreatic duct, like pseudocysts [26]. However, both MCNs and IPMNs have high cyst fluid CEA levels, produced by the inner lining of columnar epithelium in mucinous cysts [26]. Genetic analysis of the cyst fluid also shows mutations, like KRAS and GNAS (less commonly in MCNs [26].

Although most PCLs can be distinguished using known imaging characteristics and EUS–FNA, it is reasonable to assume small cysts with unremarkable features as mucinous cysts and manage as such [27].

## 3. Current Guidelines for PLC Management

Since most PDACs develop from mucinous pancreatic cysts, most guidelines focus on IPMNs and MCNs for identifying high-risk features and management [26]. A comparison of different guidelines is summarized in Table 1 for target cystic lesions and invasive recommendations and surveillance of mucinous pancreatic cysts without high-risk features.

### 3.1. History and Development of PCL Guidelines

The first guideline for the management of IPMNs and MCNs was the International Association of Pancreatology’s (IAP) 2006 Sendai Guidelines [45]. These guidelines were the first to introduce high-risk/worrisome features, like cyst size ≥ 3 cm, pancreatic duct dilatation and mural nodules, and the first to recommend follow-up for smaller cysts [45]. These guidelines were later updated as Fukuoka guidelines in 2012, and included EUS–FNA for further assessment, together with clearer distinction of high-risk/worrisome features [46]. The same guidelines were revised in 2017, focusing on only IPMNs and including use of novel biomarkers, like CEA for cyst fluid analysis and genetic analysis for KRAS–GNAS mutations [11]. The most recent iteration of IAP guidelines is that of the 2023 Kyoto Guidelines, which has included new-onset diabetes as a high-risk feature and recommendations for stopping follow-up for small, stable cysts [12].

The European Study Group (ESG) guidelines were first published in 2013, adapting similar high-risk/worrisome features, and recommending EUS–FNA for cyst fluid analysis for markers like CEA [47]. ESC guidelines were later revised in 2018, incorporating a personalized approach based on patient age and comorbidities, new recommendations, like genetic mutation assessment of cyst fluid, and discontinuing follow-up for stable cysts [48].

The 2015 American Gastroenterological Association (AGA) guidelines, first among the American guidelines, were more conservative than their counterparts at the time [49]. These guidelines aimed to reduce overtreatment by recommending avoidance of testing for asymptomatic cysts < 3 cm, discontinuing surveillance for stable cysts after 5 years, and targeting MD–IPMNs, SPNs and cNETs as requiring surgical intervention [49]. The 2017 American College of Radiology (ACR) guidelines provided an imaging-based approach to risk stratification of incidental PCLs [50]. The 2018 American College of Gastroenterology (ACG) guidelines [30] incorporated patient symptoms, high-risk/worrisome features, and molecular tests, balancing AGA’s conservative approach with 2017 Fukuoka’s invasive approach.

Finally, the 2020 International Cancer of the Pancreas Screening Consortium (CAPS) guidelines [51] mainly discuss the management of high-risk individuals with a family history of PDAC or genetic mutations related to PDAC. However, they also include recommendations for incidental PCLs found during PDAC surveillance, since PCLs in these patients have a higher risk of developing malignancy [51].

### 3.2. Overlapping Radiological Risk Factors in PCL Guidelines

Apart from the 2015 AGA guidelines, which require at least two high-risk features to indicate EUS-guided FNA or surgery, other guidelines suggest that the presence of at least one high-risk or worrisome feature is sufficient for such procedures [12,30,49,51,52]. These guidelines commonly overlap in identifying high-risk or worrisome radiological features of pancreatic cysts, including a cyst size of ≥3 cm (cm), main pancreatic duct (PD) dilatation, mural nodules, solid components within the cyst, and malignant cytology from EUS-guided FNA [26].

A meta-analysis by Anand and colleagues [52] identified a cyst size ≥ 3cm as a strong predictive factor for malignancy, with an odds ratio of 62.4. Consequently, most guidelines regard a cyst size of ≥ 3cm (≥4 cm for ESG) as a worrisome feature [12,30,48,49,50]. In the same study [52], a PD diameter of ≥6 mm was associated with an odds ratio of 7.27 for developing malignancy. Most guidelines consider a main PD diameter of ≥10 mm as a high-risk feature [12,48,50,51], while the 2018 ACG guidelines [30] consider a main PD diameter of ≥5 mm, and the 2015 AGA guidelines [49] consider any main PD dilatation as high-risk. For guidelines specifying worrisome features, the 2018 ACG and 2023 IAP guidelines consider a main PD size of 5–9 mm worrisome [12,48], whereas the 2017 ACR guidelines use a PD caliber of ≥7 mm.

Solid components and mural nodules within the pancreatic parenchyma are high-risk or worrisome features often grouped together [26]. Anand and colleagues [52] found that the presence of a mural nodule had an odds ratio of 9.3 for malignancy. All guidelines recognize the presence of a solid enhancing component as a high-risk feature [12,30,49,50,51]. The presence of a mural nodule is deemed high-risk in the 2018 ACG [30] and 2020 CAPS [51] guidelines, while the 2018 ESG [48] and the 2023 IAP [12] guidelines consider an enhancing mural nodule size of <5 mm as worrisome and ≥5 mm as high-risk. The 2017 ACR [50] guidelines regard non-enhancing mural nodules of any size as worrisome.

Other overlapping high-risk or worrisome features across recent guidelines include cyst size growth rate [12,30,48], thickened or enhanced cyst walls [12,50,51], and abrupt changes in PD caliber with distal or upstream pancreatic parenchymal atrophy [12,30,51]. Given that some PCLs can present multifocally, it is critical to assess each cyst individually for high-risk or worrisome features [26,50]. Additionally, evaluating the entire pancreatic parenchyma for solid components and other lesions is essential, as patients with IPMNs have an increased risk of developing PDAC from another location within the pancreas, referred to as a “field defect” [30]. IPMN-related parenchymal changes may also indicate an elevated risk for pancreatic cancer [34].

### 3.3. Other Common Risk Factors in PCL Guidelines

Clinical and laboratory findings, such as loss of appetite, weight loss, jaundice, abdominal or back pain associated with pancreatitis, and elevated hepatobiliary and pancreatic enzymes, often indicate high-risk or malignant cysts [12,30,48,50,51]. More recent guidelines have incorporated recommendations for EUS-guided FNA if the patient has elevated levels of carbohydrate antigen 19-9 (Ca 19-9), and new onset or acute exacerbation of diabetes mellitus [12,30,48,51]. Studies have shown that, among newly diagnosed diabetes mellitus patients over the age of 50, 1% will develop PDAC within three years [53]. Therefore, these parameters are crucial during the diagnostic work-up of pancreatic cysts.

FNA of cyst fluid helps identify mucinous cysts by their mucin content and increased levels of CEA > 192 ng/mL [54]. It is also useful for ruling in benign cysts; CEA levels < 5 ng/mL suggest pseudocysts or SCAs, and an amylase level of >250 IU/L is more indicative of pseudocysts [55]. However, CEA levels are not reliable for distinguishing between benign and precancerous cysts [48], nor are they helpful in identifying high-grade dysplasia or PDAC within mucinous cysts [30]. In these cases, cyst fluid cytology can be useful for identifying PDACs, although low cellularity remains a significant limitation [30]. The wide variation in criteria for defining risk and need for surgery emphasizes the lack of clear-cut consensus and limited accuracy of all current clinical and imaging features.

### 3.4. Future Development of Guidelines

Emerging diagnostic tools for malignant cysts are currently under investigation. Recent studies have shown that mucinous cysts have lower cyst fluid glucose levels on FNA compared to non-mucinous cysts [56,57]. A study by Ribaldone and colleagues [58] found that a cyst fluid glucose concentration of <50 mg/dL was more sensitive for diagnosing mucinous cysts than a CEA level of >192 ng/mL, with both CEA and glucose levels demonstrating high specificity for mucinous cysts. A meta-analysis of 8 studies with 609 pancreatic cysts [57] compared cyst fluid glucose to cyst fluid CEA and found higher pooled sensitivity (91% versus 56%) and higher diagnostic accuracy (94% versus 85%) for cyst fluid glucose, without a significant difference in specificity (86% versus 96%).

Next-generation sequencing of cyst fluid for KRAS and GNAS mutations has shown high specificity for mucinous cysts [59,60,61]. Other mutations, such as TP53, PIK3CA and/or PTEN, have been associated with IPMNs with advanced neoplasia [59]. Including these advancements in the guidelines, alongside further progress in medical knowledge and new diagnostic tools, enhances the accuracy of these guidelines in detecting malignant cysts. However, these recent advancements often require invasive procedures (EUS–FNA), which are currently limited to cysts with worrisome or high-risk features.

### 3.5. Differing Views in PCL Guidelines

Although these guidelines outline several overlapping risk features for pancreatic cysts that necessitate invasive interventions, they differ significantly in management strategies, and their effectiveness continues to be investigated [62,63]. The 2015 AGA [49] and the 2018 ACG [30] guidelines concur on surgery for all MD-IPMNs and SPNs. The 2018 ESG guidelines [48] recommend surgery for all SPNs and cNETs larger than 2 cm, while the 2015 AGA [49] guidelines recommend surgery for all cNETs regardless of size.

The primary challenge lies in identifying IPMNs and MCNs that will progress to PDACs and determining which patients would benefit most from surgical intervention. Recommendations for MCNs generally align with those for IPMNs, and the two are often grouped together in most guidelines [26]. The 2015 AGA guidelines [49] stipulate the need for both pancreatic duct dilatation and a solid component to recommend surgery, while other guidelines consider a single high-risk feature sufficient for surgery consideration [12,30,48,50,51].

Suspicious (high-grade dysplasia) or positive (malignant) cytology on EUS-guided FNA is deemed sufficient for surgical intervention across most guidelines [12,30,48,49,51], except for the 2017 ACR guidelines [50], which focus solely on radiologic features. The potential benefit of long-term surveillance is also addressed in the guidelines. A multicenter study by Kwong and colleagues [64] found that only 1% of all patients with pancreatic cysts developed PDAC after five years of follow-up. The AGA guidelines estimate that the likelihood of an incidental cyst on MRI being a ductal cancer is 17 in 100,000, and only about 17% of patients undergoing pancreatic resection for IPMNs had high-grade dysplasia [49]. Other studies report that 25% of patients who had surgery for presumed mucinous cysts had no malignant potential, and up to 78% of patients undergoing surgery to remove BD-IPMN had no high-grade dysplasia or PDAC [65,66].

These findings highlight differences in guideline recommendations for surgery and long-term surveillance. Lekkerkerker and colleagues [67] compared the effectiveness of three guidelines in 119 patients who underwent pancreatic surgery for PCLs. They found that, while no patient with high-grade lesions or malignancy would have been missed according to the 2012 IAP [46] or the 2013 ESG [47] guidelines, four patients (12%) would have been missed according to the 2015 AGA guidelines [63]. However, the same study reported that the indication for surgery was justified in 59% of patients according to AGA guidelines, compared to 53% with 2013 ESG and 54% with 2012 IAP guidelines [67].

A Monte Carlo simulation model study with a cohort of 10,000 patients by Lobo and colleagues [63] compared the 2017 IAP guidelines [11] with the 2015 AGA guidelines [49]. The study found that, while both guidelines were similar in terms of PCL management-related deaths and quality-adjusted life years, the 2017 IAP guidelines resulted in more surgeries, surgery-related deaths, cross-sectional imaging studies, and a higher cost per cancer identified [63]. Conversely, the AGA guidelines were associated with more missed cancers and more cancer-related deaths but had a similar number of all-cause deaths at a lower cost [63].

These studies show that, while guidelines that recommend long-term surveillance do aid in early detection of PDACs, they result in unnecessary interventions with harmful results and higher costs while providing benefit only to those limited cysts that are or are likely to become malignant. Recent guidelines recommend lifelong surveillance but also include criteria for stopping surveillance in stable pancreatic cysts, and in patients who are unfit for surgery [12,30,48,50]. Therefore, the management of pancreatic cysts depends not only on the properties of the cyst, but also on the long-term benefits and preferences of the patient.

## 4. Imaging (Radiologic) Tests for PCL Evaluation

PCLs are usually detected incidentally in cross-sectional imaging. For cysts with high-risk or worrisome features, EUS–FNA is recommended by all guidelines [26]. For cysts without features suggestive of high malignant risk, current methods of surveillance are MRI, magnetic resonance cholangiopancreatography (MRCP), CT, and EUS [12,68]. The advantages and disadvantages of different radiological tests are summarized in Table 2.

### 4.1. Current Guideline Recommendations for Non-Invasive Imaging

While some guidelines, like the 2018 ESG [48], recommend MRI as the preferred imaging modality for PCL surveillance, the 2017 ACR [50], the 2018 ACG [30], and the 2023 IAP Kyoto [12] guidelines also include CT as an alternative method. The ESG guidelines have reported that the accuracy of identifying PCL type was 40–81% with abdominal CT and 40–95% with MRI/MRCP [48]. They highlight several advantages of MRI/MRCP over CT, such as higher sensitivity for detecting multiple cysts, mural nodules, internal septations, communication with the main pancreatic duct, and isoattenuation PDACs [48,68]. The ACG guidelines report that the accuracy of MRI or MRCP was 40–50% for determining cyst type and 55–76% for differentiating benign cysts from malignant ones, with similar accuracies for CT and EUS without FNA [30]. The ACG guidelines refer to a study by Jones and colleagues [28], which found that the accuracy of CT for identifying benign cysts from malignant ones was 71–80%; however, MRI/MRCP has 96% sensitivity for IPMNs, likely due to its higher accuracy for identifying communication with the pancreatic duct [28]. Another crucial point of consideration is the effects of long-term repeated exposure to ionizing radiation for patients undergoing surveillance with CT [48]. However, CT is useful for showing cysts with calcifications, like SCAs with central calcification or MCNs with peripheral calcifications [48,50]. CT is also useful for assessing vascular invasion, metastases, stage, and postoperative recurrence of pancreatic cancer [48]. Some studies have also shown that CT, MRI and EUS have similar accuracy in diagnosing IPMNs with high-risk features [69]. A study by Min and colleagues has shown that MRI and CT were comparable for diagnosing high-risk lesions (75.4% versus 73.7%) [70].

### 4.2. Current Guideline Recommendations for EUS

Recent guidelines include EUS as an alternating method to MRI during surveillance [30,48,51], or as the next step for further investigation of PCLs with high-risk features. EUS has a higher sensitivity than CT or MRI for IPMNs < 1 cm in size due to its higher resolution, and it can detect deeply localized tumors, lymph node metastases, and vascular invasion, with the significant advantage of the ability to obtain fluid and tissue samples with FNA biopsy [30,71]. A study by Du and colleagues [72] also found that EUS had a higher diagnostic sensitivity and accuracy for differentiating malignant PCLs than CT or MRI. A review study by Tirkes and colleagues [73] reported that EUS alone (without FNA of cyst fluid) had 65–96% accuracy for diagnosing benign cysts from malignant ones, similar to MRI and CT. However, some studies have also demonstrated that CT with 3D reconstructions or MRI with MRCP is equal to EUS in establishing the connection to the main pancreatic duct [28,74]. Some studies also report that EUS has a lower specificity than CT or MRI [30]. Other important limitations of EUS are its dependence on the investigator’s expertise and the risk of adverse events [30,48]. Still, combining CT or MRI with EUS may be considered for high-risk cysts due to increased accuracy in detecting neoplasms [30].

### 4.3. Future Development for Imaging

Recently, new EUS modalities have been investigated for PCLs. Contrast-enhanced EUS (CE-EUS) is useful for assessing vascularity within the cysts and septations, and the 2018 ESG guidelines recommend it for further evaluation of suspected mural nodules, as vascular flow clearly distinguished these from similar appearing mucous globules [48]. Recent studies report higher accuracy in diagnosing PCLs compared to conventional imaging modalities [75]. EUS-guided needle-based confocal laser endomicroscopy (nCLE) is a method that uses a microscopic probe loaded onto the FNA needle and advanced into the cyst to visualize and assess its internal lining for cyst characteristics and lesions [76]. It has been demonstrated as a practical method for differentiating IPMNs, other PCLs, and high-risk features, like papillary epithelial thickness and darkness [15,77]. Although nCLE results could be reported differently according to the interpretation of the user, a study by Machicado and colleagues [78] using machine learning models has demonstrated high sensitivity and specificity (83% and 88%, respectively) for identifying advanced neoplasia in IPMNs. EUS-guided micro forceps (MFB) involves a micro-forceps that is passed through the 19-gauge EUS needle to collect tissue samples from PCLs, with a higher yield than conventional EUS-guided FNA [15]. In a study by Tacelli and colleagues [79], EUS-guided MFB outperformed cytology regarding diagnostic yield for mucinous cysts, with a sensitivity of 88.6% and specificity of 94.7% for mucinous cysts. However, the high risk of adverse events, like acute pancreatitis and intra-cystic bleeding, limits EUS-guided-MFB’s inclusion in daily practice [18,79]. Although novel methods improve pancreatic cyst differentiation, application of DL methods could provide automated differentiation and better risk stratification of PCLs.

## 5. Artificial Intelligence for PCL Management

Artificial intelligence, specifically DL and radiomics, presents a transformative opportunity for PCL management. From improved diagnoses and risk stratification to personalized treatment plans and potentially more precise minimally invasive procedures, DL can significantly improve patient care and outcomes in this complex clinical area. Similarly, radiomics, a rapidly evolving field that leverages quantitative features extracted from medical images, offers a promising approach to the refining of PCL management. In the following sub-sections, we summarize how DL and radiomics are utilized for PCL management: diagnosis (classification) and segmentation (organ and pathology boundary identification) (See Figure 1).

Deep learning: DL has emerged as a powerful tool with the potential to significantly support PCL management across several key areas. First, DL algorithms may improve PCL diagnosis and risk stratification. Automated PCL detection and segmentation in medical images (CT scans, MRIs) can alleviate radiologist workload and potentially enhance diagnostic accuracy. Furthermore, DL models, trained on extensive datasets encompassing PCL images and patient data, can aid in the estimation of malignancy risk, guiding clinicians toward informed decisions regarding surgery versus surveillance. DL can extract subtle features from images that might be overlooked by human observation, leading to more refined diagnoses and risk assessments. Secondly, DL paves the way for personalized treatment planning for PCLs. By incorporating not only imaging data, but also patient demographics, medical history, and even genetic information, DL models can create personalized risk profiles, enabling the development of more tailored treatment strategies. Additionally, DL-powered decision support systems can assist clinicians in selecting the optimal course of action for each patient based on their unique risk profile. Finally, DL is promising for advancing minimally invasive techniques in PCL management. Real-time image guidance systems for procedures like EUS-guided FNA, powered by DL, could improve needle placement accuracy, and potentially reduce complications. Similarly, DL may contribute to the development of robots that can perform minimally invasive PCL surgery with enhanced precision, potentially leading to faster recovery times.

Radiomics: Radiomics encompasses the extraction of quantitative imaging features that extend beyond visual assessment, aiming to develop actionable databases from radiologic images, and is increasingly being utilized for diagnosis, prognosis, prediction, and therapeutic response assessment in oncology. These radiomics “hand-crafted” features include (1) semantic radiologist derived assessments of the tumor including spiculations, size of the tumor along several axes, (2) shape-based features that quantitative measure regular or irregular tumor boundary changes based on their 3D topology, and (3) intra-tumoral heterogeneity measures, including gray-level features, which investigate pixel level textural differences to characterize the heterogeneity of a tumor. The extraction and analysis of these features are fully automated, enabling high-throughput processing. Briefly, radiomics transforms medical images into analyzable data via the following workflow: delineation of regions of interest (for example, segmentation of the region of interest), extraction and quantification of features, and development of diagnostic/predictive/prognostic models. Specifically, in the context of pancreatic disease, by autonomously learning intricate patterns within the data, radiomics algorithms can potentially identify subtle abnormalities associated with pancreatic cancer, leading to earlier and more precise diagnoses.

Radiomics and DL applied CT, MRI, and EUS-based studies for PCL management are summarized below.

### 5.1. CT-Based Studies (Segmentation and Diagnosis)

Accurate segmentation of the pancreas and classification of PCLs on CT scans can be extremely beneficial for optimizing patient management in the era of personalized medicine. Precise segmentation enables the quantification of PCL volume, a key factor influencing malignancy risk. Moreover, robust classification of PCLs into benign and malignant categories is essential for guiding clinical decision-making. By distinguishing benign lesions suitable for surveillance from malignant PCLs requiring surgical intervention, CT-based analysis can minimize unnecessary procedures and enhance patient outcomes. This technology holds promise for improving the management of indeterminate PCLs, where current diagnostic methods often lack clarity. Overall, developing accurate and efficient CT-based segmentation and classification tools can revolutionize PCL diagnosis and treatment, leading to a more personalized approach to pancreatic care.

#### 5.1.1. Segmentation of Pancreas from CT

Recent advancements in DL have enabled automatic pancreas segmentation in CT images (See Table 3) [80,81,82,83,84,85,86,87,88]. Several notable datasets are utilized for pancreas segmentation, including the NIH pancreas segmentation dataset, which consists of 82 scans, and the MSD CT segmentation datasets, which comprise 420 scans for tumor cases [89,90]. Other datasets, such as AMOS or AbdomenCT-1K [91,92,93], which are not explicitly designed for pancreas segmentation, also contain pancreas annotations.

Over the past decade, various DL methods have been proposed for CT segmentation, ranging from 2D to 3D approaches and traditional convolutional neural networks (CNNs) to recent self-attention-based segmentation methods. 2D CNNs are favored for pancreas segmentation due to their computational efficiency and lower data requirements. However, they struggle to capture local spatial features in the complex pancreas of the structures. In contrast, 3D CNNs excel in capturing spatial details across CT slices, which is essential for accurate pancreas segmentation and for reducing false positives and negatives, thereby enhancing overall performance. However, 3D CNNs are inherently more complex and require greater computational resources.

The pioneering work, DeepOrgan, introduced a multi-level deep CNN to represent the anatomical structure of the pancreas, achieving a Dice coefficient of 71.8% on the NIH pancreas segmentation task [94]. To enhance focus on specific pancreas regions, the Attention U-Net approach incorporates Attention Gates [80]. These gates emphasize critical image areas for precise delineation, improving the focus on the pancreas and yielding more accurate segmentations, with a Dice coefficient of 83.1% on the same NIH Pancreas segmentation dataset, without extensive architectural modifications. This method is adaptable, easily integrated into any CNN architecture, and imposes minimal computational overhead, making it widely adopted in medical image segmentation.

The recent landmark nnUNet, i.e., nothing new U-Net, updates the U-Net architecture by incorporating prior architectural considerations, and addressing the challenge of designing effective CNNs for medical image segmentation, a task that demands considerable expertise and time [95]. nnUNet automates the design process through optimization algorithms that identify optimal CNN architectures for specific segmentation tasks. Its self-designing capability, enhanced by dense data augmentation, has demonstrated robust performance across various medical segments, including pancreas segmentation, achieving Dice scores of 82.14% for the pancreas body and 54.28% for pancreatic cancer on the MSD dataset.

Recently, self-attention-based Transformers have revolutionized pancreas segmentation by providing long-distance relationship modeling capabilities, which are particularly beneficial given the elongated structure of the pancreas. Zhang demonstrated that linear self-attention approximations enable effective 3D segmentation using transformers, achieving Dice coefficients of 85.5% on NIH pancreas datasets and 83.3% on the Medical Segmentation Decathlon [96] datasets. These results are comparable or superior to state-of-the-art methods. Sample segmentation results based on this state-of-the-art method are shown in Figure 4.

Lastly, Qiu and colleagues [97] introduced a two-stage approach to apply transformers to localized pancreas regions, utilizing a residual transformer block for multi-scale feature extraction and addressing pancreas location variability. They incorporated dual convolutional down-sampling to refine shape and size features, achieving a Dice coefficient of 86.25% on a publicly available NIH dataset [97].

#### 5.1.2. CT-Based PCL Diagnosis with AI and Radiomics

While segmentation is used for recognizing and isolating the target organ, classification is required to understand the type and clinical significance of the lesion. Classification studies on PCLs mainly focus on distinguishing the type of cyst (Table 4). Dimitriev and colleagues [98] trained a DL model using CT imaging features of PCLs of 134 patients to classify them into the four most common types (IPMNs, MPNs, SCNs, and SPNs). Using a Bayesian combination of random forest and CNN classification, they achieved an overall classification accuracy of 83.6% [98]. Another study by Liang and colleagues [99] trained support vector machine (SVM) and logistic regression models to differentiate PCLs with data obtained from CT images and found an area under the curve (AUC) of 0.92 for diagnosing SCNs, and an AUC of 0.97 for differentiating between MCNs and IPMNs. A similar study by Chu and colleagues [100] trained a radiomics-based random forest model with features from preoperative CTs and demographics (age and gender) of 214 patients and achieved an AUC of 0.940 for distinguishing between five types of PCLs (IPMNs, MCNs, SPNs, SCNs, and cNETs), which was higher than that of experienced radiologists (AUC: 0.895) [100]. Some studies have also evaluated algorithms trained on labeled datasets for risk prediction. A radio-genomic-based model by Permuth and colleagues [101] combined radiomics features on CT imaging and microRNA genomics data of 38 patients to identify malignant IPMNs, achieving an AUC of 0.92, sensitivity of 83%, and a specificity of 89% while using only worrisome features defined by guidelines achieved an AUC of 0.54. Hanania and colleagues [102] evaluated a logistic regression model for IPMN risk prediction in a cohort of 53 patients, achieving an AUC of 0.96, a sensitivity of 97%, and a specificity of 88%, with a lower false positive rate compared to the 2012 IAP (Fukuoka) guidelines (5% versus 36% respectively). Small sample sizes were a key limitation in both studies [101,102]. Chakraborty and colleagues [103] developed random forest and SVM models using radiologic features and five clinical variables (age, gender, cyst size, presence of solid component, and symptoms) from 103 patients to categorize IPMNs into low or high-risk, confirmed with resection and histopathologic evaluation. Their algorithm achieved an AUC of 0.77 with radiologic features alone, with increased performance (AUC: 0.81) after including clinical variables [103].

Most of the previously mentioned research (Table 4) is commonly carried out by manual annotations, which are time-consuming, expensive, and include inter-observer variations. Some studies have tried to overcome this problem with automated or semi-automated methods. For example, Si and colleagues [104] developed an end-to-end DL model for diagnosing pancreatic tumors without the need for manual preprocessing. They evaluated 170 PDAC patients and 17 IPMN patients in the testing dataset; their model identified IPMNs and PDACS with accuracy rates of 100% and 87.6%, respectively [104], with an average of 18.6 s per patient, which was significantly faster than the radiologist evaluation. Such time-saving approaches could be useful for clinical workflow. Natural language processing (NLP) based systems are also being investigated to streamline PCL management and follow-up. Yamashita and colleagues [117] used NLP to extract patients with PCLs and lesion measurements from large amounts of historical CT and MRI reports. Based on the radiologists’ annotations as ground truth, the model had 98.2% true-positive and 3% false-positive rates [117].

#### 5.1.3. MRI-Based Studies (Segmentation and Diagnosis)

MRI offers a valuable tool for segmenting and classifying pancreatic cystic lesions (PCLs), aiding in the development of personalized treatment strategies. Unlike CT scans, MRI excels at delineating soft tissue contrast, allowing for more accurate segmentation of PCLs and surrounding pancreatic tissue. This precise segmentation facilitates measuring features like wall thickness and diffusion characteristics, which can be crucial for differentiating benign from malignant lesions. Furthermore, advanced MRI techniques hold promise for non-invasively characterizing PCLs on a molecular level, potentially leading to earlier and more accurate diagnoses. By providing detailed anatomical and functional information, MRI-based segmentation and classification can empower clinicians to tailor treatment plans for PCLs, improving patient outcomes and reducing the need for unnecessary surgical procedures.

#### 5.1.4. Segmentation of Pancreas from MRI

Unlike the advancements in CT-based pancreas segmentation, automatic segmentation studies with MRI remain limited due to lack of publicly available datasets (Table 3). In principle, all previous methods developed for CT pancreas can be applied to MRI scans, but performance might remain questionable. The only publicly available MRI dataset is the AMOS dataset [91], which contains only 40 publicly available MRIs. Asaturyan and colleagues [86] introduced a new method to accurately segment the pancreas in MRI volumes from multiple protocols (T2-weighted, fat-suppressed) using a Hausdorff–Sine loss function to incorporate anatomical shape information. This method achieved Dice scores of 84.10 for 180 in-house T2-weighted and 85.7 for 120 fat-suppressed images [86]. Cai and colleagues [84] further improved pancreatic detection and boundary segmentation by differentiating pancreas and non-pancreas tissues using spatial intensity context and allocating semantic boundaries. Their method attained a mean Dice score of 76.1 with a standard deviation of 8.7 across 78 in-house abdominal MRI scans. Salanitri and colleagues [87] introduced a multi-decoder architecture using multiple multi-head attention mechanisms via 3D decoders, each predicting intermediate segmentation maps that combine into a detailed pancreas segmentation mask, achieving a Dice score of 77.46 on 40 in-house MRI-T2 images. Alternatively, Cai and colleagues [85] applied a recurrent convolutional neural network architecture to the segmentation challenge, achieving a Dice score of 80.5 on 79 MRI scans, displaying a different approach from attention-based strategies. Very recently, a new algorithm (PanSegNet) was developed by Zhang and colleagues [88] by combining nnUNet and a Transformer network with a new linear attention module. They achieved impressive Dice scores of 85.0% and 86.3% for T1W and T2W MRI segmentation, respectively, using a large multi-center dataset of 767 MRI scans. Notably, this study introduces the first publicly available MRI dataset for pancreas segmentation, consisting of 767 scans from 499 participants across five centers, which the authors have made available to the research community. The same study also showed state-of-the-art results for CT segmentation, evaluated over 1350 CT scans, and software was made available to the public [88].

#### 5.1.5. MRI-Based PCL Diagnosis with AI and Radiomics

Integrating AI and radiomics into MRI analysis of PCLs holds immense promise for enhancing diagnostic accuracy, standardizing interpretation, and improving patient care pathways. AI-driven MRI data analysis for PCL diagnosis was approached through DL models, particularly convolutional neural networks (CNNs), in the early years of CNNs, followed by Transformers and other joint architectures. Studies in MRI-based PCL diagnosis with AI and radiomics are summarized in Table 4. One of the earlier works was carried out by Chen and colleagues [105], who used various 3D CNNs (ResNet 18, ResNet34, ResNet52, and Inception-ResNet) to classify pancreatic tumors from MRI images. They [105] used 77 benign MRI images from 20 normal patients and 38 malignant MRI images from 20 patients and used data augmentation to create 442 benign and 421 malignant MRI images. Among those benchmarked methods, the ResNet18 model achieved an accuracy of 91% for correctly classifying lesions. Further, conventional radiomics were used for PCL management.

Cheng [106] compared CT and MRI radiomics for predicting malignant potential in IPMNs. The study [106] found that MRI radiomics features (extracted from manually delineated cysts) had higher reproducibility than CT features. Further, the MRI radiomics model using SVM achieved the best performance, with an AUC of 0.940 for differentiating malignant from benign IPMNs [106]; outperforming CT radiomics models (AUC 0.811–0.864) and a clinical/imaging model based on Fukuoka guidelines (AUC 0.764). Corral, 2019 [107] and Hussein, 2019 [22] evaluated several DL algorithms for IPMN risk assessment and compared them to two guidelines (AGA and Fukuoka guidelines). The DL algorithm had a sensitivity of 75% and specificity of 78% for detecting high-grade dysplasia or cancer; its diagnostic accuracy was comparable to the AGA and Fukuoka guidelines, with areas under the ROC curve of 0.78 for DL, 0.77 for AGA, and 0.77 for Fukuoka (*p* = 0.90) [107]. The DL method offered slightly higher sensitivity than the other approaches. It was able to classify IPMNs in just 1.82 s, compared to 5–10 min for manual radiologist review [107]. On the other hand, Cui (2021) [108], developed a radiomics nomogram to predict the grade of branch duct IPMNs (BD-IPMNs) of the pancreas preoperatively. MRI-derived radiomic features with clinical characteristics were used to construct the nomogram [108]. The final nomogram incorporating the radiomics signature with CA19-9 levels and main pancreatic duct size improved performance, with AUCs of 0.903, 0.884, and 0.876 in the three cohorts [108]. The nomogram demonstrated good calibration and clinical utility based on decision curve analysis [108]. In a similar line of study with BD-IPMNs, Flammia (2023) [110] used diffusion MRI images (ADC maps) in addition to anatomical MRI and obtained AUC: 0.80 for predicting the development of worrisome features in low-risk BD-IPMNs.

In a more recent study, Salanitri, 2022 [109] proposed using transformer neural networks to classify IPMNs by developing a method using vision transformers pre-trained on natural images and fine-tuned on MRI data. Their approach achieved 70% accuracy in classifying IPMNs as normal, low-risk, or high-risk, outperforming several convolutional neural network baselines. The transformer model showed better generalization and interpretability compared to CNNs [109]. While slightly lower than the 73% accuracy of a specialized CNN architecture from previous work, the authors note that their general transformer approach performed remarkably well on this complex medical task with limited training data [109]. Finally, in 2023, Yao and colleagues [111] presented a framework consisting of an automated pancreas segmentation, a DL classifier, a radiomics classifier, and a decision fusion mechanism. The combined DL and radiomics approach reached 81.9% accuracy in IPMN risk classification, surpassing the previous state-of-the-art performance of 61.3% [111]. The study [111] used a dataset of 246 MRI scans from five different medical centers, demonstrating the method’s effectiveness across diverse data sources. The authors in [111] also found that incorporating pancreas volume as a clinical feature improved classification performance.

All these studies suggested that radiomics and DL may improve diagnostic accuracy for assessing IPMN malignancy risk compared to conventional methods, with better efficiency. Despite the promising potential of AI and radiomics in PCL diagnosis, current research in this area remains limited. Challenges include the need for large, diverse, and well-annotated MRI datasets of PCLs, which are essential for training and validating AI models. Additionally, the heterogeneity of MRI acquisition protocols across institutions poses a significant hurdle to developing generalizable models. Prospective, multi-center studies are needed to rigorously evaluate the performance of AI and radiomic approaches in real-world clinical settings, comparing them against current diagnostic standards and assessing their impact on patient outcomes. While significant work remains, developing AI and radiomic tools for MRI-based PCL diagnosis represents a promising frontier in pancreatic imaging and precision medicine.

#### 5.1.6. EUS-Based Studies (Segmentation and Diagnosis)

As with CT and MRI, multiple groups have applied ML/DL methods to EUS imaging with promising results for diagnosis. Pure segmentation studies are extremely limited in EUS; therefore, Table 4 includes both segmentation and diagnosis work in the same table. The more relevant challenge with EUS is that of cyst diagnosis once a lesion is manually identified by the endoscopist. While promising, these also require real-time image processing to be clinically relevant, given the nature of endoscopy and the need to decide, in real time, whether fine needle aspiration should be performed.

#### 5.1.7. Auto-Segmentation of Pancreatic Cysts from EUS

Although EUS is widely used to evaluate various pancreatic diseases, developing specific neural networks designed for automatic pancreas segmentation based on EUS images or videos remains limited. The EUS images acquired depend highly on the performing physician, as the endoscopic is manually maneuvered into semi-standardized positions to image the pancreas. As such, segmentation algorithms would require real-time image processing to guide such maneuvers accurately. Tang and colleagues [75] employed UNet++, a well-established and robust network architecture, for pancreatic mass segmentation, achieving an accuracy of 92.3%. Machicado and colleagues [78] introduced a mask region-based convolutional neural network (CNN) for EUS video segmentation, facilitating more accurate diagnoses of pancreatic diseases. Similarly, Zhang and colleagues [118] applied UNet++ to 247 EUS videos for training and 44 for testing, targeting the pancreas and abdominal aorta, and achieved a Dice score of 83.6 on the testing data. While various studies have focused on EUS-based pancreas and tumor segmentation, the only study explicitly specifically addressing pancreatic cystic lesion (PCL) segmentation is by Oh and colleagues [112]. They developed an Attention U-net model for automated PCL segmentation in 111 patients, reporting a Dice score of 0.794, Intersection over Union (IoU) score of 0.741, pixel accuracy of 0.983, sensitivity of 0.797, specificity of 0.991, and the highest recall at IoU > 0.50 and IoU > 0.75. Despite these advancements, research in accurate pancreas segmentation for EUS images remains relatively unexplored, indicating a significant opportunity for further development and improvement.

#### 5.1.8. EUS-Based PCL Diagnosis (Classification)

Nguon and colleagues [114] developed DL cyst classification tools for EUS. While EUS-based PCL segmentation studies are lacking, EUS-based classification has been attempted more frequently. Kuwahara and colleagues [113] used a ResNet-50 model for 3970 images from 50 patients to differentiate benign and malignant IPMNs. After training the model with 10-fold cross-validation, their [113] model achieved sensitivity, specificity, PPV, NPV, and accuracy of 81.5%, 90.1%, 86.5%, 86.2%, and 86.2%, with a threshold of 0.49. They [113] considered the mean prediction value of all images for each patient as the mean malignant probabilities of the patient. In patient-level prediction, 0.41 was used as a threshold; the sensitivity, specificity, PPV, NPV, and accuracy were 95.7%, 92.6%, 91.7%, 96.2%, and 94.0%, respectively, which was greater than human preoperative diagnosis [113].

Machicado and colleagues [78] introduced a three-stage diagnosis strategy by using CNN-based binary classifiers to predict high-grade dysplasia/carcinoma and low/intermediate dysplasia of IPMNs. EUS-nCLE videos from 35 patients were converted to 15027 frames for building the CNN model, and 5-cross validation was employed [78]. In the first stage, the CNN VGG-16 network was trained for segmentation of identifying the papillary structures, and the model obtained a sensitivity of 73.6%, a specificity of 82.8%, and an AUC of 87.3% [78]. In the second stage, frames with papillary structures were investigated with two different algorithms: a segmentation-based model (SBM) for detecting papillary epithelial thickness and darkness and a holistic-based model (HBM) that classified IPMNs according to risk [78]. SBM achieved accuracy, sensitivity, and specificity of 82.9%, 83.3% and 82.4%, respectively, while HBM achieved accuracy, sensitivity, and specificity of 85.7%, 83.3% and 88.2%, respectively.

Nguon and colleagues [114] used a CNN model based on the ResNet-50 backbone, using pre-trained weights on the ImageNet dataset to build the model in order to differentiate between MCNs and SCNs. Authors in [113,114] developed two distinct fine-tuned approaches for transfer learning and evaluated models in single-ROI and multi-ROI test sets. In a single-ROI test set, the model fine-tuned with the ResNet-FC method showed accuracy, sensitivity, specificity, and AUC of 62.29%, 60.67%, 63.30%, and 0.68, respectively [114]. In contrast, the model fine-tuned in ResNet-Conv+FC method reached a higher accuracy, sensitivity, specificity, and AUC of 82.76%, 81.46%, 84.36%, and 0.88, respectively [114]. In a multi-ROI test set, a model fine-tuned with the ResNet-FC method achieved accuracy, sensitivity, specificity, and AUC of 62.56%, 60.35%, 64.14%, and 0.67, respectively [114]. In contrast, the model fine-tuned with ResNet-Conv+FC showed accuracy, sensitivity, specificity, and AUC of 80.00%, 76.06%, 84.55%, and 0.84, respectively [114].

Vilas-Boas and colleagues [115] trained a CNN model, employing a Xception model and ImageNet pre-trained weights during training to 5505 EUS video images extracted from 28 distinct patients to differentiate mucinous cysts from non-mucinous ones [115]. Authors achieved an AUC for discrimination of 1, accuracy of 98.5%, sensitivity of 98.3%, specificity of 98.9%, PPV of 99.5%, and NPV of 96.4% [115]. Most recently, Schulz and colleagues [116] trained a CNN model with 3355 EUS images from 43 patients with histologically proven IPMN to differentiate between benign and malignant IPMNs and achieved an accuracy of 0.996 in predicting histologic outcomes.

## 6. Discussion and Future Prospects

While AI/DL has ignited a revolution in medical image analysis with significant improvements in pancreatic cancer and pre-cancerous diagnosis, it is crucial to acknowledge that we are not yet at the finish line. Detecting pancreatic cancer at its earliest and most treatable stages remains a significant challenge. There are three primary reasons behind this challenge.

(1)Data scarcity: Obtaining high-quality, labeled datasets for training DL models is a hurdle. Limited data can restrict the model’s ability to generalize effectively and accurately identify subtle early-stage cysts and predict their progress into pancreatic cancer.(2)Tumor Heterogeneity: Pancreatic cysts exhibit a high degree of variability in their appearance. DL models require vast amounts of diverse data to learn the nuances that differentiate high-risk and low-risk cysts, as well as high-risk cysts versus pancreatic cancerous tumors.(3)Biological Complexity: Pancreatic cancer is a complex disease driven by intricate biological processes. DL models, while powerful, primarily focus on image features. Integrating additional clinical data, like genetic information, could enhance their ability to detect early signs of the disease.

There are also other challenges beyond data scarcity, tumor heterogeneity, and biological complexity. A fundamental concern is the need for robust AI models that can maintain consistent performance across diverse clinical settings and patient populations. This challenge is compounded by the need for models to maintain accuracy even when encountering unusual or rare PCL presentations that may not have been well-represented in training datasets. The computational demands of AI systems present another significant barrier to widespread implementation. Advanced deep learning algorithms, particularly those processing complex three-dimensional imaging data from CT and MRI scans, require substantial computational resources both for training and real-time clinical application. These requirements translate into significant infrastructure costs for healthcare institutions, including investments in high-performance computing systems, data storage solutions, and specialized hardware accelerators. Clinical implementation faces additional barriers that extend beyond technical considerations. Healthcare institutions must navigate complex regulatory requirements for implementing AI-based medical devices, ensuring compliance with data privacy regulations, and establishing protocols for regular system validation and monitoring.

Despite these challenges, ongoing research is actively addressing these limitations. Larger, more comprehensive datasets are being compiled, and scientists are exploring methods to incorporate multi-modal data (combining imaging with genetic information) into DL models. The journey towards early detection of pancreatic cancer through AI and PCLs is ongoing, and the potential for significant breakthroughs remains very promising.

However, realizing the full potential of DL in PCL management requires overcoming certain challenges. The availability of large, high-quality datasets encompassing images and patient data is crucial for training effective DL models. Balancing data privacy and security concerns is paramount in this context. Additionally, ensuring generalizability of models trained on data from specific institutions is essential. This necessitates the use of diverse datasets and robust model development techniques. Finally, fostering trust with clinicians requires understanding how DL models reach their conclusions. Research efforts are underway to develop methods for explaining model predictions and addressing this critical aspect.

Explainable (or interpretable) AI, in this manner, is critical for clinical decision support. As AI systems become more complex, there is a growing need for explainable AI in clinical settings. In modern healthcare, interpretable AI serves as a crucial bridge between advanced machine learning capabilities and practical medical applications. At its core, interpretable AI enables medical professionals to understand and validate the reasoning behind AI-generated recommendations, establishing a foundation of trust and accountability in medical decision-making. This transparency allows doctors to verify whether AI conclusions align with established medical knowledge and identify the specific factors that led to particular diagnoses or treatment suggestions. The ability to interpret AI decisions also plays a vital role in clinical validation and regulatory compliance. Medical institutions can thoroughly audit AI systems to ensure they meet rigorous clinical standards, while regulators can effectively evaluate their safety and reliability. This interpretability creates a clear documentation trail for legal and compliance purposes, which is essential in the highly regulated healthcare environment. The continuous improvement and clinical translation of medical AI systems heavily depends on their interpretability. When researchers and development teams can understand the decision-making processes of their AI models, they can identify and correct errors, refine algorithms, and enhance overall performance. This iterative improvement process ensures that medical AI systems become increasingly reliable and effective over time. Perhaps most importantly, interpretable AI significantly enhances patient care and engagement. When medical professionals can clearly explain how AI supports their diagnoses and treatment plans, patients gain a better understanding of their healthcare decisions. This transparency helps build trust between patients and healthcare providers, leading to better compliance with treatment plans and ultimately improved health outcomes.

While explainable AI is highly studied nowadays for various applications in healthcare, the research effort in PLC management is limited from an explainability perspective. Future research should focus on developing interpretable models that provide clinicians with clear rationales for their predictions or recommendations. This transparency is crucial for building trust among healthcare providers and patients, facilitating the integration of AI tools into clinical workflows. Explainable AI could also offer insights into decision-making, potentially uncovering new biomarkers or imaging features relevant to pancreatic cyst management.

Multimodal DL refers to artificial intelligence models and techniques that can simultaneously process and learn from multiple types of data or sensory inputs, like how humans integrate information from various senses. This approach combines different modalities of data, such as text, images, audio, video, and sometimes even tactile or other sensory data. In healthcare, the future of multimodal DL for pancreas disease research holds immense promise for improved diagnosis, prognosis, and treatment stratification. By integrating diverse data sources, like imaging modalities (CT, MRI), genomics, and clinical variables, multimodal DL can capture the complex interplay of factors that influence disease progression. This comprehensive approach can identify novel disease subtypes, predict treatment response with greater accuracy, and personalize care for patients with pancreatic diseases. Early detection of pancreatic malignancies, a significant challenge, could be significantly improved through multimodal DL-based analysis of subtle abnormalities in multiparametric MRI scans. Similarly, multimodal DL-powered integration of genetic and imaging data could guide the development of targeted therapies and predict the risk of treatment complications. As multimodal DL algorithms become more sophisticated and datasets grow, their role in unraveling the biological underpinnings of pancreatic disease will become increasingly crucial, paving the way for the discovery of new therapeutic targets and the optimization of clinical decision-making.

As for another future research perspective, there is a pressing need for large-scale, longitudinal studies to train AI models on the temporal evolution of pancreatic cysts. Such research could enable the development of predictive models that forecast cyst growth, malignant transformation, or resolution over time. These predictive capabilities would be invaluable for clinicians in determining optimal surveillance intervals and identifying patients who might benefit from early intervention. Moreover, this approach could help reduce unnecessary procedures and alleviate patient anxiety associated with frequent follow-ups.

While many AI models show promise in controlled research settings, their real-world performance and clinical impact remain largely unexplored. Future studies should focus on implementing AI tools in diverse clinical environments that will assess their performance across different patient populations and healthcare systems. This research should evaluate diagnostic accuracy and metrics, such as cost-effectiveness, patient outcomes, and changes in clinical decision-making. Such studies are essential for understanding the true value of AI in pancreatic cyst management and identifying areas for improvement.

Since AI systems hold unprecedented power over human life, death, and well-being in medical settings, the ethics of AI should be carefully examined. Ethical implications in healthcare extend far beyond mere technical transparency, touching on fundamental aspects of human dignity, autonomy, and justice in medical care [119]. When AI systems make decisions that affect patient health outcomes, the ability to interpret and explain these decisions becomes an ethical imperative, ensuring that patients maintain their autonomy through informed consent and understanding of their treatment options. This interpretability also addresses critical concerns about algorithmic bias and fairness, allowing healthcare providers to identify and mitigate potential discriminatory patterns that could disadvantage certain demographic groups. The ethical dimension also encompasses broader societal considerations, such as equitable access to AI-enhanced healthcare services and the responsible use of patient data in training these systems, making interpretability a cornerstone of ethical AI deployment in medicine.

Finally, an exciting avenue for future research lies in leveraging AI to develop personalized treatment plans for patients with pancreatic cysts. This could involve creating models that can simulate different treatment scenarios and predict outcomes based on a patient’s unique characteristics. Such systems could help clinicians and patients make more informed decisions about pursuing watchful waiting, surgical intervention, or other management strategies. Additionally, AI could potentially identify patient subgroups that are more likely to benefit from specific treatments, paving the way for more targeted and effective therapeutic approaches in pancreatic cyst management. Most recent breeds of AI techniques, called Foundational Models, i.e., large-scale AI systems trained on vast and diverse datasets, are poised to revolutionize medical applications in the near future [119]. These models, exemplified by systems like GPT (Generative Pre-Trained Transformer), have demonstrated remarkable capabilities in understanding and generating human-like text, as well as processing and analyzing complex data across various domains. In healthcare, foundational models hold promise for enhancing diagnostic accuracy, predicting treatment outcomes, and personalizing patient care. As research progresses, we anticipate the emergence of more sophisticated methods leveraging these models, including advanced techniques, such as prompt tuning and engineering. These approaches will enable more precise and context-aware interactions with medical data, potentially transforming tasks ranging from clinical decision support to medical research and education. One immediate application of a foundational model will be a new clinical decision support for pancreatic cyst management, but this time not only with clinical and imaging data but also analysis of guidelines, medical literature, genomic and other patient data to provide more evidence-based recommendations tailored to individual cases.

## 7. Conclusions

In conclusion, managing pancreatic cystic lesions remains a significant clinical challenge, balancing the need for early detection of malignant potential against overtreatment of benign lesions. While conventional radiographic approaches have been the mainstay of pancreatic cystic lesion evaluation, they are limited by subjective interpretation and variable accuracy. Integrating artificial intelligence and advanced radiology imaging into pancreatic cystic lesion management offers a promising path forward, potentially enhancing diagnostic precision, improving risk stratification, and personalizing patient care strategies. AI-driven techniques for lesion segmentation and radiomic analysis could provide more objective and reproducible assessments, potentially revolutionizing clinical decision-making. However, the full realization of AI’s potential in pancreatic cystic lesion management requires further validation through rigorous, multi-center prospective studies. As this field evolves, MRI is being used more often than other modalities and AI is becoming an invaluable tool in the clinician’s arsenal, ultimately improving diagnostic accuracy in PCLs and patient outcomes in the challenging landscape of pancreatic cystic lesions.

## Figures and Tables

**Figure 1 cancers-16-04268-f001:**
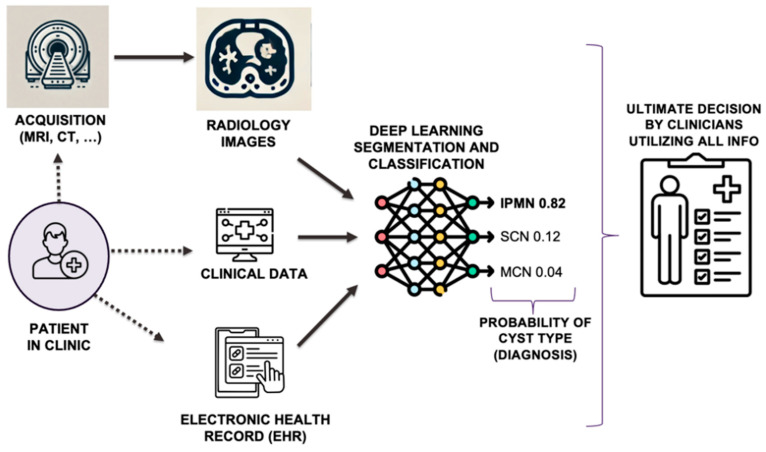
An example of AI workflow utilizing multi-modal data for diagnosing pancreatic diseases (cancer and cysts). Patient clinical and electronic health record information and imaging data are used within a deep learning segmentation (for images) and prediction algorithm to diagnose suspicious activity in the pancreas. The clinicians make the ultimate decision once AI-derived and other results are presented to them.

**Figure 2 cancers-16-04268-f002:**
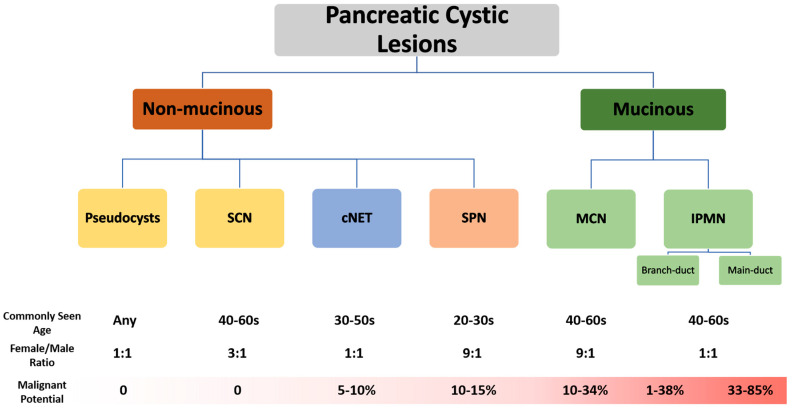
Six most common types of Pancreatic Cystic Lesions. **SCN**: serous cystic neoplasm; **cNET**: cystic pancreatic neuroendocrine tumor; **SPN**: solid-pseudopapillary neoplasm; **MCN**: mucinous cystic neoplasm; **IPMN**: intraductal papillary mucinous neoplasm.

**Figure 3 cancers-16-04268-f003:**
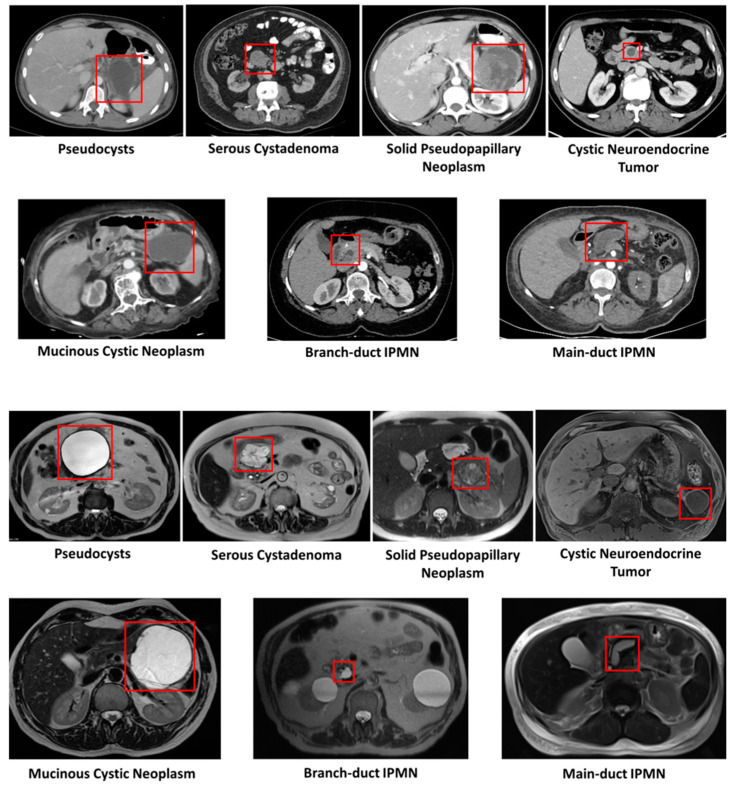
CT (first two rows) and MRI (third and fourth rows) examples of Pancreatic Cysts.

**Figure 4 cancers-16-04268-f004:**
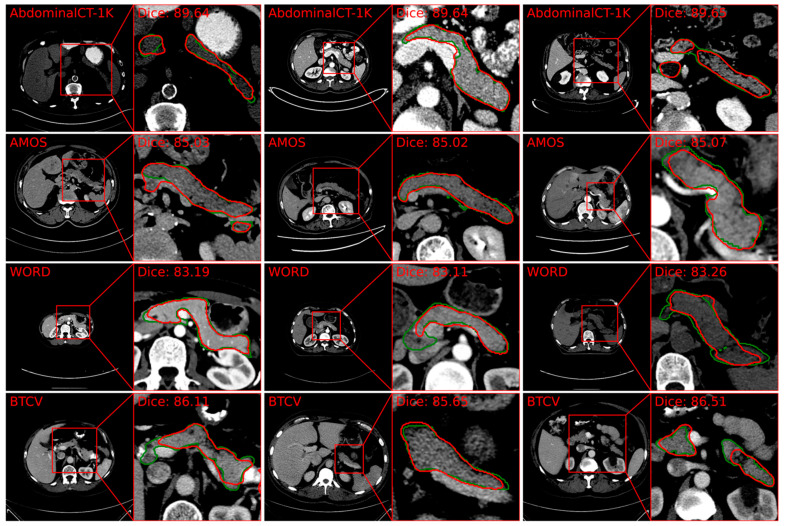
First row: CT Pancreas segmentation performance on various CT datasets. Second row: Pancreas segmentation results from MRI T1 (**left**) and T2 (**right**) cases. Green: predicted, red: ground truths.

**Table 1 cancers-16-04268-t001:** Comparison of different guidelines’ target cystic lesions, invasive recommendations, and surveillance of mucinous pancreatic cysts without high-risk/worrisome features.

Guidelines	2015 AGA *	2017 ACR	2018 ACG	2018 ESG	2020 CAPS	2023 IAP (Kyoto)
**Target lesions**	Only asymptomatic cysts and MD-IPMNs are excluded *	PCLs	PCLs (mainly IPMNs and MCNs)	PCLs (mainly IPMNs and MCNs)	All pancreatic abnormalities	IPMNs
**High-risk features**	-Cyst size ≥ 3 cm -Dilated main PD -Solid component	-Main PD caliber ≥ 10 mm -Enhancing solid component -Obstructive jaundice with a cyst in the head of pancreas	-Cyst size ≥ 3 cm -Main PD > 5 mm -Change in main PD caliber with upstream pancreatic atrophy -Mural nodule or a solid component -Growth in size ≥ 3 mm/year -Acute pancreatitis due to cyst -Jaundice due to cyst -Elevated Ca-19-9 -Suspicious or positive cytology †	-Main PD ≥ 10 mm -Enhancing mural nodule ≥ 5 mm -Solid component -Jaundice (cyst related)	- Main PD dilatation ≥ 10 mm -Mural nodule -Enhanced solid component -Thickened or enhanced walls - Abrupt change in PD caliber with distal pancreatic atrophy -Symptoms present (pancreatitis, jaundice, pain) -Positive cytology †	-Main PD ≥ 10 mm -Enhancing mural nodule ≥ 5 mm or a solid component -Obstructive jaundice with a cyst in the head of pancreas -Suspicious or positive cytology †
**Worrisome features**	N/D	-Cyst size ≥ 3 cm -Main PD caliber ≥ 7 mm -Thickened/enhancing cyst wall -Non-enhancing mural nodule	N/D	-Cyst size ≥ 4 cm -Main PD 5–9.9 mm -Enhancing mural nodules < 5 mm -Growth rate ≥ 5 mm/year -Acute pancreatitis due to IPMN -Ca-19-9 > 37 U/mL (in the absence of jaundice) -New-onset diabetes mellitus	N/D	-Cyst size ≥ 3 cm -Main PD 5–9 mm -Abrupt change in PD caliber with distal pancreatic atrophy -Enhancing mural nodule < 5 mm -Thickened/enhancing cyst wall -Rapid growth (> 2.5 mm/year) -Pancreatitis due to cyst -Lymphadenopathy -Ca-19-9 elevation -New onset or acute exacerbation of diabetes mellitus within one year ‡
**Indications for EUS/FNA**	≥2 high-risk features	High-risk and/or worrisome features	≥1 high-risk feature -New-onset/worsening diabetes mellitus	High-risk and/or worrisome features (not recommended if diagnosis is already established)	Mural nodule, solid component, or main PD dilatation	Presence of high-risk features ‡ and/or worrisome features
**Indications for surgery**	Cyst with both a solid component and PD dilatation -Concerning features in cytology † -MD-IPMNs, solid-pseudopapillary neoplasms, and cystic neuroendocrine tumors *	High-risk and/or worrisome features	-Main PD involvement -Mural nodule -Suspicious or positive cytology † -All Solid-pseudopapillary neoplasms	Absolute indications: - ≥ 1 high-risk feature -Suspicious or positive cytology † Relative indications: -Patients without comorbidity and ≥1 worrisome feature -Patients with co-morbidities and ≥ 2 worrisome features -All solid-pseudopapillary neoplasms -Cystic pancreatic neuroendocrine tumors > 2 cm	Cysts with high-risk features	- ≥1 high-risk feature -Suspicious or positive cytology † - ≥2 worrisome features ‡ -Repeated acute pancreatitis worsening patient’s quality of life ‡ -Young, fit for surgery
Comparison of different guidelines’ recommendations for surveillance of mucinous pancreatic cysts without high-risk/worrisome features.						
**Imaging methods**	MRI	MRI or CT	MRI/MRCP and/or EUS	MRI and/or EUS (along with serum CA-19-9 and clinical evaluation)	MRI/MRCP and/or EUS (along with serum CA-19-9, fasting serum glucose and/or HbA1c evaluation)	MRI/MRCP, CT EUS for further investigation
**Intervals**	<3 cm lesions: In 1 year, then every 2 years ×2	-Yearly for < 1.5 cm lesions: Yearly 1.5–2.5 cm lesions: Every 6 months × 4, lengthen if stable -> 2.5 cm every year for 10 years, for patients that are > 80 years, every 2 years	-Cysts <1 cm: MRI every 2 years × 4, lengthen if stable -Cysts 1–2 cm: MRI yearly ×3, if stable every 2 years × 4 and then lengthen -Cysts 2–3 cm: MRI or EUS in 6–12 months for 3 years, if stable every 2 years × 4 and then lengthen -> 3 cm lesions: Alternate between MRI and EUS in 6 months × 3 years, if stable every year × 4 and then lengthen -Increase in cysts size and/or new-onset diabetes: MRI or EUS within 6 months; if stable, MRI in 1 year and then return to base surveillance	-Cysts <1.5 cm: Yearly ×3, if stable then every 2 years -Cysts >1.5 cm: Every 6 months ×2, if stable then yearly	- <3 cm lesions: Every year - >3 cm lesions: Every 6 months	- <2 cm lesions: First in 6 months, then every 18 months, if stable ‡ 2–3 cm lesions: 6 months ×2, then every year, if stable ‡ - >3 cm lesions: Every 6 months or with 1–6 month-interval according to estimated risk ‡
**Duration**	Stop after 5 years if stable	Stop after 10 years if stable or at age 80 Stop if the patient is unfit for surgery	Lifelong, review after age 75 Stop if the patient is unfit for surgery	Lifelong, stop if the patient is unwilling or unfit for surgery	Lifelong	Lifelong Consider stopping for <2 cm cysts without high-risk/worrisome features that are stable after 5 years ‡ Stop if the patient is unfit for surgery or has a life expectance of <10 years

* 2015 AGA guidelines only focus on asymptomatic cysts. MD-IPMNs without side branch involvement; solid-pseudopapillary neoplasms, cystic degenerations of adenocarcinomas, and neuroendocrine tumors are not discussed since they are less challenging to identify, and their accepted management is surgical resection; † on EUS and FNA; suspicious cytology: high-grade dysplasia and positive cytology: malignancy; ‡ changes in the 2023 IAP Kyoto guidelines compared to the 2017 IAP Fukuoka Guidelines. ACG, American College of Gastroenterology; ACR, American College of Radiology; AGA, American Gastrointestinal Association; CAPS, Cancer of the Pancreas Screening; Ca-19-9, carbohydrate antigen 19-9; CT, computed tomography; EUS, endoscopic ultrasound; ESG, European Study Group; FNA, fine-needle aspiration; IAP, International Association of Pancreatology; IPMN, intraductal papillary mucinous neoplasm; MCN, mucinous cystic neoplasm; MD, main duct; MRCP, magnetic resonance cholangiopancreatography; MRI, magnetic resonance imaging, N/D, Not defined; PD, pancreatic duct.

**Table 2 cancers-16-04268-t002:** Comparison of different imaging tests for PCLs.

Modality	Advantages	Disadvantages
**US**	Widely available Cost-effective Real-time imaging Doppler capability No radiation	Limited resolution Operator-dependent Poor visualization in obese patients Limited sensitivity for small or complex cysts
**CT**	Widely available Fast acquisition High resolution	Radiation exposure Limited soft-tissue contrast Contrast medium risk (allergy, kidney injury)
**MRI/MRCP**	Excellent soft-tissue contrast Better cyst characterization Detailed ductal evaluation (MRCP) No radiation	Higher cost Longer scan times Contraindication (intracardiac devices, implants) Limited availability
**EUS**	High resolution Biopsy capability (FNA) Close, detailed imaging	Invasive (required sedation) Procedural risks (infection, bleeding, pancreatitis) Operator-dependent Higher cost
**PET**	Functional imaging for metabolic activity Useful adjunct for malignancy suspicion	Limited specificity Radiation exposure Higher cost Limited availability

CT, computed tomography; EUS, endoscopic ultrasound; FNA, fine-needle aspiration; MRCP, magnetic resonance cholangiopancreatography; MRI, magnetic resonance imaging; PET, positron emission tomography; US, ultrasound.

**Table 3 cancers-16-04268-t003:** (Selected) representative CT and MRI-based pancreas segmentation studies.

**Study (CTs)**	**ML Model**	**Study Type**	**Number of Patients**	**Performance (Dice *) %**
Oktay, 2018 [80]	AttentionUNet	Pancreas body	82 NIH data	83.1 ± 3.8%
Ning, 2018 [81]	Deep Qnetwork	Pancreas body	82 NIH data	86.9 ± 4.9%
Zhao, 2019 [82]	Two-stage 3D model	Pancreas body	82 NIH data	86.0 ± 4.5%
Lim, 2022 [83]	Four individual 3D pancreas segmentation networks	Pancreas body	1006 in-house CT scans	84.2%
**Study (MRIs)**	**ML Model**	**Study Type**	**Number of Patients**	**Performance (Dice) %**
Cai, 2016 [84]	Graph-based decision fusion with CNN	Pancreas body	78 in-house T1 MRI scans	76.1 ± 8.7%
Cai, 2017 [85]	Recurrent neural contextual learning	Pancreas body	79 in-house T1 MRI scans	80.5 ± 6.7%
Asaturyan, 2019 [86]	A Hausdorff-Sine loss function was used to incorporate anatomical shape information.	Pancreas body	180 in-house T2-weighted, 120 fat-suppressed MRI scans	84.1% for T2-weighted, 85.7% for fat-suppressed MRI scans
Proietto Salanitri, 2021 [87]	Multiheaded decoder structure	Pancreas body	40 In-house T2 scans	77.5 ± 8.6%
Zhang [88]	Linear Transformer with nnU-Net (PanSegNet)	Pancreas body	1350 CT scans and 767 T1 and T2 (made publicly available)	88.3% for CT, 85.0% for MRI T1, 86.3% for MRI T2.

* Dice score: Higher is better. CNN, Convolutional neural network; CT, computed tomography; MRI, magnetic resonance imaging, NIH, National Institutes of Health.

**Table 4 cancers-16-04268-t004:** CT, MRI, and EUS-based PCL classification studies.

**Study (CTs)**	**ML Model**	**Classification Type**	**Number of Patients/Images**	**Performance**
Permuth, 2016 [101]	Combined radiomic features on CT imaging and microRNA genomic data	Identifying malignant IPMNs	38	AUC: 0.92 Sensitivity: 83% Specificity: 89%
Hanania, 2016 [102]	Logistic regression model	IPMN risk prediction	53	AUC: 0.96 Sensitivity: 97% Specificity: 88%
Dimitriev, 2017 [98]	Bayesian combination of the random forest, CNN classification	Classification of PCLs	134	Overall accuracy: 83.6%
Chakraborty, 2018 [103]	Random forest and SVM models using radiologic features and five clinical variables	Categorizing IPMNs into low or high-risk	103	AUC: 0.77 (radiologic features alone) AUC: 0.81(after including clinical variables)
Si, 2021 [104]	Fully end-to-end DL model	Diagnosing pancreatic tumors without manual preprocessing	170 PDAC,17 IPMN	Accuracy. 100% (IPMNs) and 87.6% (PDACs) Processing Speed: 18.6 s per patient
Liang, 2022 [99]	SVM, logistic regression models	Differentiating PCLs	193	AUC: 0.92 (Diagnosing SCNs) AUC:0.97 (differentiating between MCNs and IPMNs)
Chu, 2022 [100]	Radiomics-based random forest model	Differentiating PCLs	214	AUC: 0.94
**Study (MRI)**	**ML Model**	**Study Type**	**Number of Patients/Images**	**Performance**
Chen, 2018 [105]	3D CNNs (ResNet18, ResNet34, ResNet52, Inception-ResNet)	Pancreatic cancer classification	40 (20 normal, 20 pancreatic cancer), 863 MRI images	ResNet18 Accuracy: 91%
Cheng, 2018 [106]	Logistic regression, SVM	Comparison of CT and MRI for predicting malignant IPMNs	60 IPMNs (37 malign, 23 benign)	AUC: 0.811–964 AUC: 0.94 for MRI
Corral, 2019 [107]	CNN	IPMN classification	139 patients	Sensitivity: 75% Specificity: 78%
Hussein, 2019 [22]	Supervised and unsupervised deep learning	IPMN classification	171 MRI images (38 normal, 133 IPMNs)	Sensitivity: 59% Specificity: 69% Unsupervised accuracy: 84.2%, but with limited specificity: 46.5%.
Cui, 2021 [108]	Radiomics	BD-IPMN Classification	202 patients	AUCs of 0.903, 0.884, and 0.876 in the three cohorts.
Proietto Salanitri, 2022 [109]	Vision Transformers	IPMN Classification	139 MRI images	Accuracy: 70%
Flammia, 2023 [110]	Radiomics	Classification of High-risk BD-IPMNs	50 MRI images	AUC: 0.80
Yao, 2023 [111]	Transformers and Radiomics	IPMN Classification	246 MRI images	Accuracy: 81.9%
**Study (EUS)**	**ML Model**	**Study Type**	**Number of Patients**	**Performance**
Oh, 2021 [112]	CNN (Attention U-net)	PCL Segmentation	111	Dice * score of 0.794 IoU score of 0.741 Accuracy: 98.3%
Kuwahara, 2019 [113]	CNN (ResNet50)	Differentiation of benign and malignant IPMNs	50	Sensitivity: 95.7% Specificity: 92.6% Accuracy: 94.0% (Accuracy of human was 56% and accuracy of a mural nodule presence was 68%)
Machicado, 2021 [78]	2 CNN algorithms SBM (segmentation-based model) and HBM (holistic-based model) for risk stratification	Segmentation of papillary epithelial thickness and darkness (SBM) and Risk Stratification of IPMNs with EUS-nCLE	35 (18 IPMNs with high-grade dysplasia or carcinoma, 17 IPMNs with low or intermediate-grade dysplasia)	Accuracy SBM: 82.9%, HBM 85.7% Sensitivity SBM: 83.3%, HBM: 83.3% Specificity SBM: 82.4%, HBM: 88.2%
Nguon, 2021 [114]	CNN (ResNet50)	Differentiation between MCNs and SCNs	109 (59 MCN, 49 SCN)	AUC of 0.88 Accuracy: 82.8% Sensitivity: 81.5% Specificity: 84.4%
Villa-Boas, 2022 [115]	CNN (Xception model trained on ImageNet)	Classification of PCLs into mucinous (IPMNs, MCNs) and non-mucinous (Pseudocysts, SCNs) types	28	AUC: 1 Accuracy: 98.5% Sensitivity: 98.3% Specificity: 98.9%
Schulz, 2023 [116]	CNN	Differentiation between benign and malignant IPMNs	43	Accuracy: 99.6%

* Dice score: Higher is better. AUC, Area under the curve; BD, branch duct; CNN, Convolutional neural network; CT: computed tomography; IoU, Intersection over union; IPMN, intraductal papillary mucinous neoplasm; MCN, mucinous cystic neoplasm; MD, main duct; MRI, magnetic resonance imaging; PCL, Pancreatic cystic lesion; SCN, Serous cystic neoplasm; SVM, support vector machine.

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
