# Peer review of "Advances for Managing Pancreatic Cystic Lesions: Integrating Imaging and AI Innovations"

_cancers, 2024, doi:10.3390/cancers16244268_

Round 1

Reviewer 1 Report

Comments and Suggestions for Authors

Advances for Managing Pancreatic Cystic Lesions: Integrating Imaging and AI Innovations

Comments: 

Introduction

    • The flow of ideas could be more gradual, starting with the general and then delving into specific details.

    • Add a brief initial definition of pancreatic cancer before discussing its impact.

    • Include context on why early diagnosis is challenging.

    • Strengthen the connection between the presented problem and the proposed solution (AI).

    • On lines 52-53: specify the time frame for the 12.5% ​​survival rate.

    • On lines 72-74: detail the explanation on why 15-20% of PCLs progress to PDAC.

    • Add a brief paragraph on current challenges in conventional imaging and mention how AI could specifically address them.

    • Improve the transition between the general discussion of pancreatic cancer and the section on AI.

PCLs

    • The introduction of this section could be better developed. It is suggested to establish the definition of PCLs, their clinical importance, and their impact on public health. 

    • Explain why these lesions represent a diagnostic challenge.

    • Figure 2 could incorporate prevalence percentages, malignancy rates, and main demographic characteristics for each type of lesion.

    • The description of pseudocysts could be better developed, detailing the mechanisms of formation, specific characteristics in imaging studies, etc.

    • Similarly, with serous cystic neoplasms (SCNs), it is suggested to detail their characteristic "honeycomb" radiological pattern and includes morphological variants (if applicable). The justification for conservative management in most cases should be explained in detail.

    • The section on mucinous cystic neoplasms (MCNs) also needs to be expanded. It is recommended to delve deeper into their association with the female gender and the presence of ovarian-type stroma.

    • IPMNs require a more detailed differentiation between their subtypes.

    • A specific section for imaging characteristics is recommended.

Current Guidelines for PLC

    • A more robust historical introduction to the development of the guidelines is suggested.

    • It would be ideal to describe the chronological evolution of the recommendations, pointing out the main changes and the factors that motivated these modifications.

    • It is suggested to present comparative data on the costs and benefits of different surveillance and management strategies.

    • It is suggested to discuss how to adapt the guidelines according to patient-specific factors, including comorbidities, preferences, and available resources. The integration of new technologies, especially AI, should be emphasized.

    • It is suggested to include a brief discussion on the probable evolution of the guidelines, considering the incorporation of new biomarkers and advanced imaging techniques.

    • Imaging Tests for PCL Assessment:

    • It is recommended to establish a clear hierarchy between the different modalities, explaining the specific advantages and limitations of each one.

    • The comparison between modalities could be more systematic. It is suggested to include a table that compares the sensitivity, specificity, predictive values, and diagnostic accuracy of each technique.

    • The section on new technologies and emerging techniques should be expanded, including information on recent advances in molecular imaging, hybrid techniques, and new applications of existing modalities.

    • It is suggested to discuss and detail decision-making based on multimodal findings.

Discussion and Future Prospects:

    • It is suggested to be more specific regarding the revolution that AI represents in the diagnosis of PCLs, considering the current limitations.

    • The main challenges could be better developed and have a clearer structure. Although three main reasons are mentioned in the text (data scarcity, tumor heterogeneity, and biological complexity), each one should be developed in more detail with specific examples.

    • It is suggested to detail the importance of the interpretability of the models for clinical acceptance.

    • It is suggested to emphasize the clinical implications, for example, it can be explained how these technological advances could change daily clinical practice.

    • It is suggested to address the ethical implications of the use of AI in the diagnosis and management of PCLs.

Conclusions

    • It is recommended to point out how the combination of advanced imaging techniques and the incorporation of AI is transforming the diagnostic landscape.

    • It is recommended to summarize how different imaging modalities, especially when complemented with AI, have improved diagnostic accuracy in PCLs.

    • It is also recommended to highlight how these advances can translate into concrete improvements in patient care.

Author Response

See attached rebuttal please.

Reviewer 2 Report

Comments and Suggestions for Authors

This review critically assesses contemporary PCL diagnostic and monitoring methodologies, delineating the characteristics of various lesions and emphasising the potential shortcomings of traditional approaches. The authors investigate the capacity of artificial intelligence (AI) to revolutionise PCL management. AI-driven methodologies, encompassing deep learning algorithms for automated pancreas and lesion segmentation, together radiomics for heterogeneity analysis, can enhance diagnostic precision and risk classification. Preliminary findings indicate that AI-driven techniques can markedly enhance patient outcomes by facilitating the earlier identification of high-risk lesions and minimising superfluous interventions for benign cysts. This review highlights that AI-driven methodologies may significantly transform PCL management, potentially enhancing pancreatic cancer prevention.

Throughout the content, the authors have successfully encapsulated the dynamic interplay between modern technologies and scientific innovation, underscoring its potential to revolutionize the field, making it more efficient and accessible. This article promises to offer valuable insights into the rapid developments in cancer research, emphasizing the importance of both technological advancement and the growing popularity of AI and imaging techniques.

In general, the paper exhibits a commendable level of writing proficiency, and the examination of capacity of artificial intelligence (AI) to revolutionise PCL management is intellectually stimulating. Tables and figures constitute a significant inclusion that has the potential to enhance the paper's citation count. I endorse the submission of this paper for publication after addressing some of my concerns:

The authors' inclusion of fundamental concepts in introduction section is missing and it should contribute to the extent body of literature (For example: https://doi.org/10.3390/cancers15092410).

Introduction should be covered the gap of the research. However, it is not well covered in this section.

Also, please mention the significance of this review to society as well as industry.

The paragraph before Table 2 should be revised, because it shows text similarity with published paper

Author Response

See attached rebuttal, please.

Reviewer 3 Report

Comments and Suggestions for Authors

Traditional diagnostic and management for PCLs, rely heavily on the expertise of the clinician, resulting in variability in interpretation and treatment decisions. This study critically examines current practices in the diagnosis and surveillance of PCLs, detailing the characteristics of various lesions and identifying the limitations of conventional methods. It also explores the transformative potential of AI in addressing these challenges. AI-driven automated image analysis and deep learning algorithms show promise in enhancing diagnostic accuracy and risk stratiffcationcan, potentially improving patient outcomes.

The review covers a wide range of relevant topics, from the biological characteristics of pancreatic cystic lesions to advancements in artiffcial intelligence AI, ensuring a holistic understanding for the researchers. The manuscrpt might be improved by addressing the following:

 1. While the scope is broad, there’s a risk of lacking depth in speciffc areas, without a clearly deffned audience (e.g., clinicians, researchers, or general readers), the review risks being either too technical or overly simplistic for certain groups.

2. The review highlights the beneffts of AI but lacks a balanced discussion of its limitations with only addressing Data scarcity, Tumor Heterogeneity, and Biological Complexity. Onther challenges such as the need for robustness model, high modeling or computational costs, and potential barriers to clinical implementation.

3. Potential concerns related to AI, such as data privacy, the need for clinician training, or the integration of AI tools into existing workffows may also be included to make this review more comprehensive.

4. Section 3 need to be reorganized:

5. line 227: This passage contains a lot of information packed into long sentences, making it difficult for readers to digest. Breaking it down into shorter paragraphs or bullet points would improve readability. Also use shorter paragraphs and organize the content into subsections may improve readability.

6. Provide a brief explanation of acronyms like MD-IPMNs, SPNs, and cNETs to ensure clarity for a broader audience.

 7. Table1: Although guideline differences are mentioned, speciffc contrasts (e.g., surgical thresholds, follow-up intervals) are not elaborated, limiting the practical value of the discussion.

8. The rationale behind certain recommendations, such as why AGA guidelines focus only on asymptomatic cysts or why CAPS targets high-risk individuals, is not explained.

 9. The impact of varying guidelines on clinical decision-making or patient outcomes is not addressed, missing an opportunity to connect the discussion to real-world practice.

10. While the evolution of guidelines is noted, there is limited discussion on how recent advancements, such as imaging technologies or AI, have inffuenced these updates.  

Author Response

See rebuttal attached, please.

Reviewer 4 Report

Comments and Suggestions for Authors

- The paper is written in a clear way and targets an application of interest for the research community. 

- Why did you focus on deep learning models ? Isn't it better to include handcrafted features models as well fo the paper to be more detailed ?

- References are sometimes old I would suggest including more recent papers (focus on including more paper of 2023 and 2024).

Author Response

See attached rebuttal, please.
